# Acetylsalicylic Acid Effect in Colorectal Cancer Taking into Account the Role of Tobacco, Alcohol and Excess Weight

**DOI:** 10.3390/ijerph20054104

**Published:** 2023-02-24

**Authors:** Didac Florensa, Jordi Mateo, Francesc Solsona, Leonardo Galván, Miquel Mesas, Ramon Piñol, Leonardo Espinosa-Leal, Pere Godoy

**Affiliations:** 1Department of Computer Engineering and Digital Design, University of Lleida, Jaume II 69, 25001 Lleida, Spain; 2Population Cancer Registry in Lleida, Santa Maria University Hospital, Av. Alcalde Rovira Roure 44, 25198 Lleida, Spain; 3Pharmacy Unit, Catalan Health Service, Av. Alcalde Rovira Roure 2, 25006 Lleida, Spain; 4SAP-Argos Department, Santa Maria University Hospital, Av. Alcalde Rovira Roure 44, 25198 Lleida, Spain; 5Catalan Health Service, Department of Health, Av. Alcalde Rovira Roure 2, 25006 Lleida, Spain; 6Graduate School and Research, Arcada University of Applied Science, Jan-Magnus Janssonin Aukio 1, 00550 Helsinki, Finland; 7Lleida Biomedical Research Institute, Av. Alcalde Rovira Roure 80, 25198 Lleida, Spain; 8CIBER Epidemiology and Public Health (CIBERESP), Health Institute Carlos III, 28029 Madrid, Spain

**Keywords:** colorectal cancer, aspirin use, excess weight, smoking, risky drinking

## Abstract

Excess weight, smoking and risky drinking are preventable risk factors for colorectal cancer (CRC). However, several studies have reported a protective association between aspirin and the risk of CRC. This article looks deeper into the relationships between risk factors and aspirin use with the risk of developing CRC. We performed a retrospective cohort study of CRC risk factors and aspirin use in persons aged >50 years in Lleida province. The participants were inhabitants with some medication prescribed between 2007 and 2016 that were linked to the Population-Based Cancer Registry to detect CRC diagnosed between 2012 and 2016. Risk factors and aspirin use were studied using the adjusted HR (aHR) with 95% confidence intervals (CI) using a Cox proportional hazard model. We included 154,715 inhabitants of Lleida (Spain) aged >50 years. Of patients with CRC, 62% were male (HR = 1.8; 95% CI: 1.6–2.2), 39.5% were overweight (HR = 2.8; 95% CI: 2.3–3.4) and 47.3% were obese (HR = 3.0; 95% CI: 2.6–3.6). Cox regression showed an association between aspirin and CRC (aHR = 0.7; 95% CI: 0.6–0.8), confirming a protective effect against CRC and an association between the risk of CRC and excess weight (aHR = 1.4; 95% CI: 1.2–1.7), smoking (aHR = 1.4; 95% CI: 1.3–1.7) and risky drinking (aHR = 1.6; 95% CI: 1.2–2.0). Our results show that aspirin use decreased the risk of CRC and corroborate the relationship between overweight, smoking and risky drinking and the risk of CRC.

## 1. Introduction

Colorectal cancer (CRC) is the third leading cause of cancer death globally and the second in Europe, and its incidence is steadily rising in developing nations [1], with nearly 520,000 new cases in Europe in 2020 [2], even though a large proportion of these case are highly preventable [3]. A study in nine European countries found that approximately 20% of CRC cases may be related to overweight, smoking and risky drinking [4]. In contrast, studies have shown that long-term aspirin use may prevent CRC [5,6].

Shaukat et al. found a direct relationship between the body mass index (BMI) and long-term CRC mortality and suggested that BMI modulation may reduce the risk of CRC mortality [7]. A recent study has shown the role of obesity and overweight in early-onset CRC, and concluded that obesity is a strong risk factor [8]. Ghazaleh Dashti et al. found an association between risky drinking and an increased risk of CRC [9]. Likewise, a study has suggested an association between passive smoking and the risk of CRC [10].

Some studies have found a protective effect of aspirin against CRC [11] and various studies have concluded that aspirin reduces the overall risk of CRC recurrence and mortality and colorectal adenomas. Ma et al. recently found that aspirin, including low-dose aspirin, reduced the risk of CRC [12]. A recent study by Zhang et al. on the effect of aspirin use for 5 and 10 years found that the continuous use of aspirin increases the protective effect on CRC [13]. A Danish study also found that the continuous use of low-dose aspirin was associated with a reduced CRC risk [14]. Some studies have shown differing results on the protective effect of aspirin due to the different designs used, the type of follow-up, the recorded aspirin consumption and the size and type of population. Although the data seem compelling, a limitation of these analyses is that they do not take into account risk factors for CRC [15]. These previous studies investigated the association between the use of aspirin and CRC, but they did not study the role played by risk factors such as tobacco smoking, alcohol or excess weight. In this study, we explore how these factors, combined with aspirin use, affect the risk of CRC in a particular society.

The objective of this study was to determine the protective effect of aspirin against CRC, taking into account the effect of other risk factors (overweight/obesity, risky drinking and smoking), in Lleida, a province in Catalonia, Spain, with a large rural population and an agri-food industry that may present specific risk factors [16,17].

## 2. Materials and Methods

### 2.1. Study Population

We conducted a retrospective cohort study of aspirin use and risk factors to analyze the impact of these factors on the risk of CRC. We carried out the study on 154,715 inhabitants of Lleida aged >50 years at the start of the study period, with data available on aspirin use from 1 January 2007 to 31 December 2016 in the Catalan Health Service (CatSalut) system. The reason for selecting this period was to ensure that those CRC cases detected in 2012 had the opportunity to be exposed to aspirin for at least five years. This population was linked to the Lleida Population-based Cancer Registry to detect CRC diagnosed between 2012 and 2016.

Data on aspirin use were obtained from the number of packages dispensed by pharmacies. Catalonia has a public health system in which medicines are dispensed in pharmacies after presenting a doctor’s prescription. Drugs administered to hospitalized patients and those prescribed by private providers are not registered in the CatSalut system, and therefore were not included in this study. The CRC cases in the sample were obtained from the Lleida Population-Based Cancer Registry, and the demographic characteristics of participants, including age and sex, were obtained from the CatSalut system. Figure 1 shows a flowchart of the study population. Initially, the pharmacy database registered 724,070 inhabitants with any prescription, although 346,365 were excluded because they did not reside in the Lleida region. Another exclusion criterion was age. We only included inhabitants aged >50 years at the start of the observed period (2007), resulting in 154,717 inhabitants. We also excluded those inhabitants who did not register the risk factors correctly, although the cases excluded were minimal.

As has been presented before, this study included different databases. To enable this linkage, it was necessary to use a personal identification code called CIP. This code is unique to each inhabitant who resides in Catalonia and permits us to identify them in the Catalan Health Service and its registers (hospitals, pharmacies or primary care centers).

### 2.2. Data Collection

Data on CRC diagnoses were obtained from the Lleida Population-Based Cancer Registry using five consecutive years of incidence data, from 2012 to 2016. This period was chosen as the available years validated by the professionals of the register. Potential CRC cases were validated by checking medical records. We used hospital and pathological anatomy records as the main information sources. Cancers were identified following the rules defined by the International Association of Cancer Registries, the International Association for Research on Cancer and the European Network of Cancer Registries.

The risk factors included were risky drinking, smoking and body mass index. This information was extracted using the eCAP software (V 20.4.3) used by primary care physicians to record all patient information, which registers information from 2001. The values of these variables at the time that this study started were obtained. Body mass index (BMI) was calculated by the weight and height of the patient using the formula $\mathrm{BMI}=\mathrm{weight}\left(\mathrm{kg}\right)/\mathrm{height}{\left(m\right)}^{2}$ and categorized as follows: 18.5–24.9 normal weight, 25–29.9 overweight and >30 obesity [18]. The ICD-10 international criteria identified risky drinking and smoking. The ICD-10 code for risky drinking is F10.2, and those for smoking are F17 (mental and behavioral disorders due to tobacco use) and Z72 (tobacco use). Risky drinking was defined as consumption of >40 g/day of alcohol in men and >24 g/day in women [19]. The Spanish Health Ministry defined these grams per day with the supervision of the WHO [20]. The software also states the date of smoking onset. Smokers were defined as exposure for >5 years before the start of the study. The reason for using a period of five years was due to a previous study that suggested that this period might increase the risk of cancer [21]. Former smokers were considered smokers because the observed points in the dataset were minimal, and adding this new category could have imbalanced the dataset. General characteristics are represented in Table 1.

### 2.3. Exposure

Aspirin was categorized according to the Anatomical Therapeutic Chemical (ATC) classification system as A01AD05 (acetylsalicylic acid) medication. The use of aspirin was evaluated based on the defined daily dose (DDD) and the milligrams (mg) accumulated dose consumed by each patient throughout the study period. The DDD is a technical unit of measurement that corresponds to the daily maintenance dose of a drug for its main indication in adults and a given route of administration. The DDDs of active ingredients are established by the WHO and published on the WHO Collaborating Center for Drug Statistics Methodology website [22,23].

Exposure was determined from computerized pharmacy data and consisted of the total DDD dispensed to an individual during the study period. For instance, if a person consumed aspirin for a while, then stopped using it and later started again, the total DDD consumed during the following period was considered. To be considered as exposed to aspirin, the total number of years of consumption had to be ≥5 years. The number of years was based on previous studies, which suggested this period as the minimum for aspirin to have a protective effect [13,24]. To consider exposure to aspirin, the minimum consumed daily was >75 mg [25,26]. The number of DDD calculated this value in mg.

### 2.4. Statistical Analysis

Descriptive analyses were performed to evaluate the association between characteristics at baseline, exposure and outcomes. Patients’ characteristics, risk factors and aspirin exposure were analyzed to determine the association with the risk of CRC. The incidence rate of CRC was calculated to each factor over a specified period. A bivariate analysis was initially used to estimate the crude hazard ratios for the association between aspirin consumption and the risk of incident CRC.

A Cox proportional hazard model was used to determine the HR and the corresponding 95% CI. The models were adjusted by sex, age, aspirin exposure, BMI, risky drinking and smoking. Subsequently, stratified models were calculated by sex.

The probability values for the statistical tests were two-tailed, and a CI that did not contain 1.0 was regarded as statistically significant. Results with wide CIs should be interpreted cautiously. All statistical analyses were performed using R (R Core Team 2019), an open-source programming language and environment for statistical analysis and graphic representation.

## 3. Results

We analyzed 154,715 inhabitants of Lleida aged >50 years, of whom 1276 (0.8%) had CRC between 2012 and 2016. The mean CRC incidence rate and the total cases by sex and age group for the five study years are shown in Figure 2a,b.

The sociodemographic information and aspirin exposure in patients with CRC (Table 2) were analyzed in the bivariate analysis.

We recorded 485 (0.8 × 1000) females and 791 (1.4 × 1000) males (HR = 1.9; 95% CI; 1.6–2.0) with CRC. Most patients were from the 60–69 years (HR = 1.8; 95% CI; 1.6–2.1) and 70–79 years age groups (HR = 2.0; 95% CI; 1.9–2.6). There were 1138 (1.2 × 1000) CRC cases without aspirin consumption and 138 with aspirin consumption (1.0 × 1000) (HR = 0.9; 95% CI; 0.8–1.1). There were 504 (1.2 × 1000) cases with overweight (HR = 2.5; 95% CI; 2.2–3.1) and 603 (1.3 × 1000) with obesity (HR = 2.7; 95% CI; 2.3–3.3), and there were 56 (2.2 × 1000) cases with risky drinking (HR = 2.1; 95% CI; 1.6–2.7), while 220 (2.0 × 1000) were smokers (HR = 2.0; 95% CI; 1.8–2.4).

Cox regression showed variations in the outcomes (Table 3). Sex, age and aspirin exposure were significantly associated with CRC. The adjusted HR (aHR) for males was 1.8 (95% CI: 1.6–2.1) and 1.8 (95% CI: 1.6–2.1) in the 60–69 years age group, 2.3 (95% CI: 1.9–2.7) in the 70–79 years age, 2.2 (95% CI: 1.8–2.6) in the 80–89 years age group and 0.2 (95% CI: 0.1–0.3) in the 90 years age group. Aspirin consumption had an aHR of 0.7 (95% CI: 0.6–0.8). The BMI also was significant. Overweight had an aHR of 1.4 (95% CI: 1.2–1.7) and obesity of 1.5 (95% CI: 1.3–1.8). Risky drinking had a significant aHR of 1.6 (95% CI: 1.2–2.0) and smoking an aHR of 1.4 (95% CI: 1.3–1.7). Figure 3 represents the adjusted hazard ratios graphically.

HRs were adjusted by gender, age, aspirin use, BMI, risky drinking and smoking.

Table 4 shows the results of the Cox regression stratified by sex. In the case of males, the results were similar to the general table. In this model, aspirin exposure remained significant (aHR: 0.7; 95% CI: 0.6–0.8), as did the BMI, risky drinking and smoking. In females, aspirin use remained significant (aHR: 0.6; 95% CI: 0.4–0.8), but, of the risk factors, only obesity remained significant (aHR: 1.4; 95% CI: 1.2–1.9). Figure 4 represents the adjusted hazard ratios graphically.

HRs were adjusted by age, aspirin use, BMI, risky drinking and smoking.

## 4. Discussion

Our results confirm the negative association between aspirin consumption and CRC independently of the other risk factors measured. Males may be at a higher risk of CRC than females but aspirin may be slightly more protective in females.

Reports support a delayed effect of aspirin on CRC [27]. A meta-analysis by Rothwell et al. examined the long-term effects of aspirin on CRC outcomes using trials of aspirin [28]. Studies on the impact of aspirin in CRC prevention have been published [6,29], although the effects of risk factors and aspirin use together have not yet been analyzed. Therefore, our findings corroborate the research in the field highlighting the protective effect of aspirin and go beyond comparing this positive effect with the negative effects caused by several risk factors.

Several recent studies have suggested an association between aspirin use and some specific cancers. Ciu et al. concluded that high-dose aspirin reduced the risk of pancreatic cancer [30]. Jacobo et al. analyzed studies on the relationship between aspirin and breast cancer [31] and concluded that aspirin consumption reduced the relative risk of breast cancer. Sieros et al. suggested that aspirin reduced the risk of esophageal cancer [32]. 

We found significant differences according to sex, suggesting that men have a higher risk of developing CRC. It has been reported that men have higher cumulative levels of smoking than women and a higher alcohol intake, which may explain the higher risk [33].

People aged between 60 and 80 years had a higher risk of CRC and the 80–89 years and 90–99 years age groups had a lower risk [34,35]. Older adults may have a differential mechanism compared with younger people. For example, aging is associated with alterations in DNA methylation, which may affect the susceptibility to cancer. The gut microbiota of older people differs from that of younger adults, which may influence drug metabolism and inflammatory processes. Genetics, underreporting and age-related physiological effects could explain the reduced risk [36].

We found some differences with respect to risk factors, such as overweight/obesity, risky drinking and smoking. Overweight represented 39.5% of total CRC cases and obesity 47.3%. Therefore, approximately 85% of patients with CRC presented excess weight, suggesting exposure to a poor diet. These results corroborated previous studies [37,38,39,40]. Excess weight is one of the most important risk factors for CRC. Individuals with a higher BMI have higher levels of chronic inflammation, and obesity may act through the gut microbiome on colorectal tumorigenesis and also promotes colorectal cancer in mice. There were notable differences in risky drinking. Patients with risky drinking had a higher risk of CRC (HR = 2.2). Meta-analyses of case–control and cohort studies suggest that high alcohol consumption might be associated with an increased risk of colorectal cancer. The epidemiological evidence has been complemented by molecular evidence on the mechanisms that could explain this association [17,41]. Similar results were obtained for smoking (HR = 2.0). The crude HR obtained also indicated this association between smoking and CRC [42]. Smoking was more closely associated with colorectal tumors that arose from non-conventional pathways, such as the serrated polyps pathway, and smoking was significantly associated with the risk of advanced serrated polyps in a screening population.

The Cox regression included all the remaining model variables, such as risk factors and aspirin exposure for CRC. Sociodemographic variables such as gender and age confirmed the correlation with CRC. Males were 1.8 times more at risk than females. This may be related to men having excess body weight and higher exposure to alcohol and smoking than women [43]. Regarding the age groups, the results confirmed that the 70–79 years age group had the highest risk, which was 2.3 times greater than the 50–59 years (ref. group) and the 69–69 years and 80–89 years age groups. Other studies found similar outcomes on the incidence and association related to CRC [44,45].

The use of aspirin for ≥5 years was significant in the Cox regression. The analysis suggested that aspirin decreased the risk of CRC. The HR was 0.7 (95% CI: 0.6–0.8), meaning that it reduced the risk of CRC by 30% [46,47]. Studies have found reductions of 20–30% [46] and 27% [47] in the risk for CRC. The risk factors were correlated with an increased risk of CRC. Overweight and obesity were significantly associated with a CRC risk 1.4 and 1.5 times higher, respectively. Obesity had a higher risk, although the HR was similar [48]. Risky drinking and smoking also had a significant HR. Risky drinking had a 1.6 times higher risk and smoking a 1.4 times higher risk. Other studies also found these associations [49,50], with a 1.3 and 1.2 times higher risk for risky drinking, respectively, and a 1.2 higher risk for smoking [50].

The Cox regression stratified by sex also obtained significant results. Men and women had similar outcomes according to age. The trends were the same as the non-stratified regression. The risk of CRC was higher in people aged 60–89 years in both sexes. The use of aspirin also maintained the association with a reduced CRC risk. Specifically, in females, aspirin could prevent CRC, in the best case, by up to 40%. A similar percentage was obtained by Cook et al. [51] in a randomized controlled trial, which showed 42% aspirin protection against CRC risk among women. These results corroborated the fact that aspirin reduces the risk of CRC in both sexes [52]. However, the results related to risk factors were significant in males. Overweight and obesity were associated with a 1.5 and 1.6 times higher risk of CRC, respectively [53]. Risky drinking and smoking were also correlated with the CRC risk. The differences between males and females may be that males more often have a poor diet and drink and smoke more than females [54]. In females, only obesity was significantly associated with an increased risk of CRC. Moreover, as a previous study concluded [55], only excess weight among men was significantly associated with increased CRC risk. In addition, the authors suggested that this risk might be reversed in obese men taking aspirin. Similarly, in our study, in the analysis stratified by normal weight and overweight/obesity, aspirin was protective against CRC in both groups, but it was only statistically significant in overweight/obese patients (Appendix A) [55]. The remaining risk factors were not related to CRC, although the HR was >1 in all of them. Individual susceptibility and the type of exposure may explain these results. Men probably have a different pattern of consumption than women and are more intensely exposed to alcohol and smoking. In addition, it seems that without the effect of aspirin, these factors are related to CRC (Appendix A).

The preventive effect of aspirin has been attributed to the inhibition of cyclooxygenase (COX), the enzyme responsible for the synthesis of prostaglandins [56,57]. COX-2 is abnormally expressed in many cancer cell lines and is involved in the processes of carcinogenesis, angiogenesis and tumor growth. Additional mechanisms of aspirin include the induction of apoptosis through COX-independent pathways. Future research should also study the role of aspirin metabolites and the role of the intestinal microbiota in cancer prevention against CRC.

Long-term aspirin is prescribed for patients with a high cardiovascular risk of non-focal continuous pain due to arthritis, and the results of this study may support this indication [58,59]. 

The study has some limitations. Firstly, some patients could buy aspirin directly in pharmacies without a doctor’s prescription, and this consumption is underreported. Second, some patients may not take the medication, even if they have purchased it at the pharmacy, and, in this case, aspirin use will be overreported. Third, although the Population-Based Cancer Registry is exhaustive, it cannot be ruled out that some cases were diagnosed in hospitals in other territories and some cases have not been correctly registered. We were unable to study the dose–response relationship between low-dose aspirin and CRC because more than 90% of aspirin use in this study was at a dose of 100 mg/day, which did not allow us to assess the highest related dose effect. Another limitation is the lack of specification of the types of CRC, such as familial polyposis or familiar cancer genetics, as a possible bias. This information was not taken into account in the register. A limitation that must be considered concerns the CRC cases diagnosed before 2012. These cases were not included because the Cancer Registry started registering cases in 2012. However, CRC cases prior to 2012 would not have had the opportunity to be exposed to risk factors or aspirin for a period of 5 or more years and would not have been recorded as incident cases in this study. Despite this, CRC is a type of cancer that can present another primary cancer a few years later; therefore, some CRC cases may be included. moreover, related to the risk factors, some bias was present due to under-reporting, although the percentage of our cases was similar to the prevalence observed in Catalonia. Finally, the impact of these excluded cases was minimal because they were younger than 50, where cancer may be unrelated to risk factors, or they were cases from other regions, and few patients had to be excluded due to a lack of information on risk factors.

The study’s strengths included the fact that data are presented on risk factors, such as excess weight, smoking and risky drinking. The study was performed with information from clinical practice, with physicians unaware of the study objectives, which avoided investigator bias.

## 5. Conclusions

This retrospective study found an association between aspirin use for ≥5 years and a reduced risk of CRC. The protective effect due to aspirin was higher in women. The results also showed an association between the risk of CRC and risk factors such as overweight, obesity, smoking and risky drinking, specifically in men. Moreover, the risk of CRC in women was significantly associated with obesity. The 70–79 and 80–89 age groups had a higher risk of CRC in men and women. Therefore, despite some limitations, such as the lack of information on food or dietary factors or some bias in the aspirin prescriptions, the results are according to the recently published literature.

In general, these results reinforce the need for public health messaging about the harmful effects of smoking, alcohol use and excess weight, and the use of aspirin to prevent CRC under prescription. They also encourage continued research into CRC to find new factors or interactions among them associated with this cancer. They also may help the health system to focus on preventing them and recommend the continuous use of aspirin under medical supervision.

## Figures and Tables

**Figure 1 ijerph-20-04104-f001:**
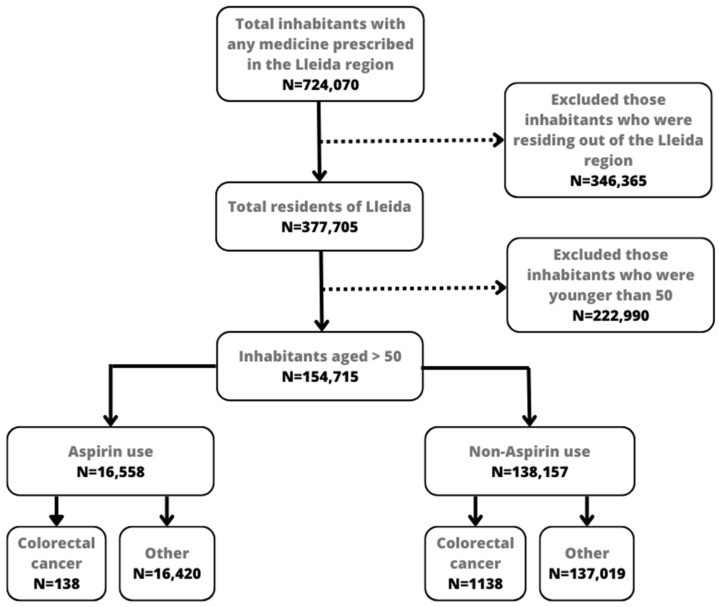
Flowchart of subjects included for the analysis.

**Figure 2 ijerph-20-04104-f002:**
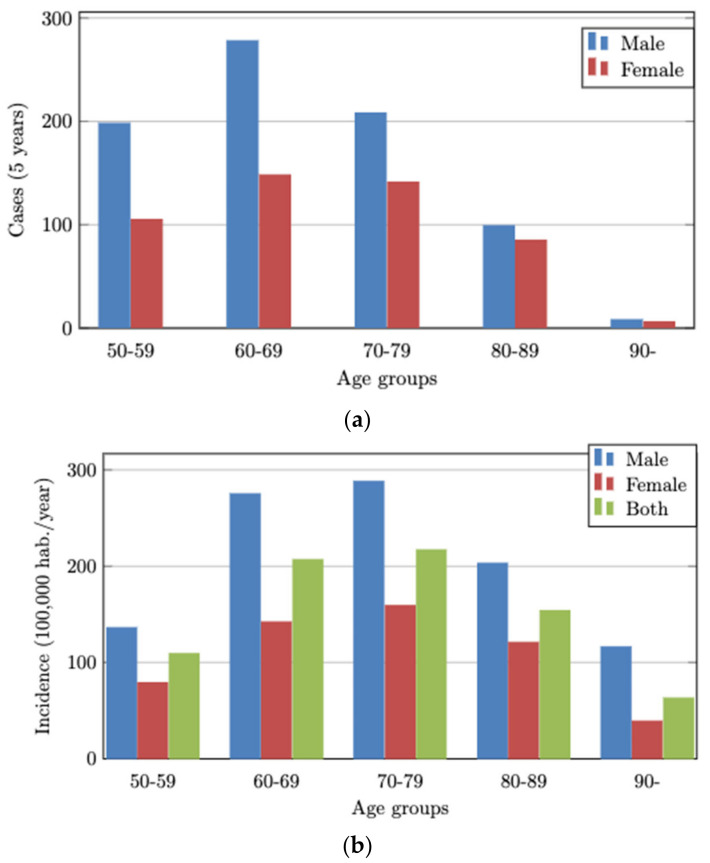
(**a**) Total CRC cases by age groups and sex for 5 years. (**b**) Mean observed CRC incidence by age groups and sex during one year.

**Figure 3 ijerph-20-04104-f003:**
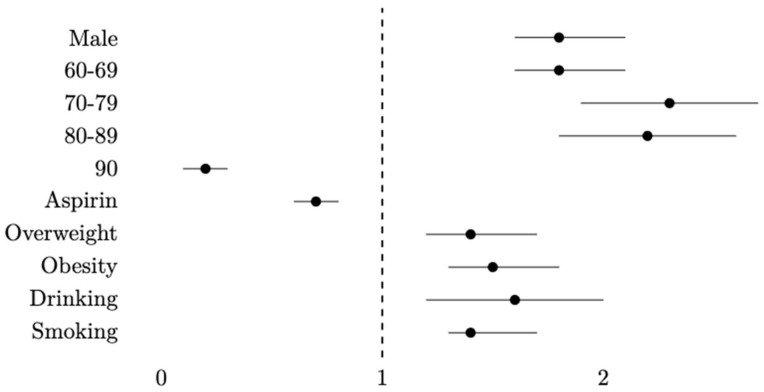
Hazard ratios by Cox regression adjusted for gender, age, aspirin use, BMI, risky drinking and tobacco smoking.

**Figure 4 ijerph-20-04104-f004:**
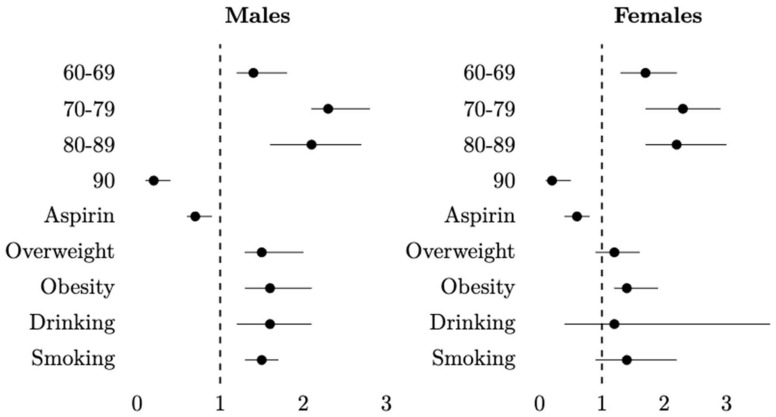
Hazard ratios by Cox regression stratified by males and females. Adjusted for age, aspirin use, BMI, risky drinking and tobacco smoking.

**Table 1 ijerph-20-04104-t001:** General characteristics of the inhabitants included in this study.

	Total		Men		Women	
	N	%	N	%	N	%
**Gender**						
Female	80,865	52.3	-	-	-	-
Male	73,850	47.7	-	-	-	-
**Age**						
[50–59)	46,454	30.0	24,080	32.6	22,374	27.7
[60–69)	35,819	23.2	17,655	23.9	18,164	22.5
[70–79)	28,138	18.2	12,842	17.4	15,296	18.9
[80–89)	23,651	15.3	10,176	13.8	13,475	16.7
[90–)	20,653	13.3	9097	12.3	11,556	14.3
**Aspirin**						
Non-use	138,157	88.1	66,695	90.3	71,462	88.4
Use	16,558	11.9	7155	9.7	9403	11.6
**Body Mass Index**						
Normal weight	47,761	30.9	21,088	28.6	26,673	33.0
Overweight	51,022	33.0	27,005	36.6	24,017	29.7
Obesity	55,932	36.1	25,757	34.9	30,175	37.3
**Risky drinking**						
No	151,323	97.8	71,997	97.5	79,326	98.1
Yes	3392	2.2	1853	2.5	1539	1.9
**Smoking**						
No	140,749	90.9	62,995	85.3	77,754	96.2
Yes	13,966	9.1	10,855	14.7	3111	3.8

**Table 2 ijerph-20-04104-t002:** Bivariate analysis with the observed years and CRC patients.

	Total		n	% (n/p-y)	Crude HR ^1^	95% CI
	Person-Year (p-y)	%				
**Gender**						
Female	639,455	53.1	485	0.8	1.0	Ref. group
Male	563,716	46.9	791	1.4	1.9	1.6–2.1
**Age**						
(50–59)	393,275	32.7	303	0.8	1.0	Ref. group
(60–69)	297,538	24.7	426	1.4	1.8	1.6–2.1
(70–79)	215,272	17.9	349	1.6	2.0	1.9–2.6
(80–89)	147,817	12.3	184	1.2	1.6	1.3–1.9
(90–)	149,269	12.4	14	0.1	0.1	0.1–0.2
**Aspirin**						
Non-use	1,068,470	88.8	1138	1.2	1.0	Ref. group
Use	134,701	11.2	138	1.0	0.9	0.8–1.1
**Body mass index**						
Normal weight	350.994	29.2	169	0.5	1.0	Ref. Group
Overweight	404.905	33.7	504	1.2	2.5	2.2–3.1
Obesity	447,272	37.2	603	1.3	2.7	2.3–3.3
**Risky drinking**						
No	1,177,736	97.9	1220	1.0	1.0	Ref. Group
Yes	25,435	2.1	56	2.2	2.1	1.6–2.7
**Smoking**						
No	1,094,891	91.0	1056	1.0	1.0	Ref. Group
Yes	108,280	9.0	220	2.0	2.0	1.8–2.4

^1^ Hazard ratio.

**Table 3 ijerph-20-04104-t003:** Multivariate analysis—Cox regression.

	Adjusted Hazard Ratio (aHR); 95% CI ^1^	*p*-Value
Female	-	Ref. Group
Male	1.8 (1.6–2.1)	<0.001
(50–59)	-	Ref. Group
(60–69)	1.8 (1.6–2.1)	<0.001
(70–79)	2.3 (1.9–2.7)	<0.001
(80–89)	2.2 (1.8–2.6)	0.007
(90-)	0.2 (0.1–0.3)	<0.001
Aspirin use	0.7 (0.6–0.8)	0.006
Normal weight	-	Ref. Group
Overweight	1.4 (1.2–1.7)	<0.001
Obesity	1.5 (1.3–1.8)	<0.001
Risky drinking	1.6 (1.2–2.0)	0.006
Smoking	1.4 (1.3–1.7)	<0.001

^1^ Confidence interval.

**Table 4 ijerph-20-04104-t004:** Adjusted Cox regression model stratified by men and women.

	Men		Women	
	Adjusted Hazard Ratio (aHR); 95% CI ^1^	*p*-Value	Adjusted Hazard Ratio (aHR); 95% CI ^1^	*p*-Value
(50–59)	-	Ref. Group	-	Ref. Group ^2^
(60–69)	1.9 (1.7–2.3)	<0.001	1.7 (1.3–2.2)	<0.001
(70–79)	2.3 (1.9–2.8)	<0.001	2.3 (1.7–2.9)	<0.001
(80–89)	2.1 (1.6–2.7)	<0.001	2.2 (1.7–3.0)	<0.001
(90–)	0.2 (0.1–0.4)	<0.001	0.2 (0.1–0.5)	<0.001
Aspirin use	0.7 (0.6–0.9)	0.005	0.6 (0.4–0.8)	0.005
Normal weight	-	Ref. Group ^2^	-	Ref. Group ^2^
Overweight	1.5 (1.2–2.0)	<0.001	1.2 (0.9–1.6)	0.1
Obesity	1.6 (1.3–2.1)	<0.001	1.4 (1.2–1.9)	0.004
Risky drinking	1.6 (1.2–2.1)	0.001	1.2 (0.4–3.7)	0.7
Smoking	1.5 (1.3–1.7)	<0.001	1.4 (0.9–2.2)	0.1

^1^ Confidence interval. ^2^ Reference group.

## Data Availability

The dataset is available from the corresponding author upon reasonable request.

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
