# Peer review of "Acetylsalicylic Acid Effect in Colorectal Cancer Taking into Account the Role of Tobacco, Alcohol and Excess Weight"

_ijerph, 2023, doi:10.3390/ijerph20054104_

Round 1

Reviewer 1 Report

Comment: Thank you for inviting me to review the paper entitled ' Acetylsalicylic acid effect in colorectal cancer taking into account the role of tobacco, alcohol and excess weight'. This paper reports a study on protective effect of aspirin against colorectal cancer, considering the effect of other risk factors weight, alcohol consumption and smoking in Lleida province in Catalonia. The topic is interesting. However, further improvement of the quality is needed. Following are some detailed comments / questions that might be useful for the improvement.

Comment 1

In the introduction, please clarify the novelty and contribution of this study. It is good to give readers a clear picture at the beginning. That will help highlight the merits of the study.

Comment 2

In chapter 2.1. Study population, please clearly specify inclusion and exclusion criteria and quantify the number and characteristics of excluded cases/inhabitants as well. Please also comment in the paper what kind of impact could excluded cases have on the results of the study.

Figure 1 shows a flowchart of subjects included in the analysis. Currently it is not clear what first two squares represent in the flowchart, who are these subjects/numbers.

 In the flowchart you are using terms “CRC diagnosed” and “Non-CRC diagnosed”. I would advise you to reframe the term “Non-CRC diagnosed”. Use of current form could be understood as subjects diagnosed with other, non- CRC, cancers as you use word “diagnosed”.

Comment 3

In chapter 2.2. Data collection line 86 you specified using 5 consecutive years of incidence data, from 2012 to 2016. Could you please add a rationale for this choice? Also, in chapter 2.1. lines 69 – 70 you stated that your study is based on data available from 1st January 2007 to 31st December 2016. Could you please explain how you handled in your study the individuals diagnosed with CRC cancer before the year 2012?

Comment 4

Lines 98 – 99: Risky drinking was defined as consumption of >40 grams/day of alcohol and >24 grams/day in 99 women. In this sentence “in men” is missing. Please also add rationale why you decided to use these limits for drinking.

Lines 100 – 103: “Smokers were defined as exposure for > 5 years before the start of the study.” Please also add rationale for this definition of exposure to smoking.

Comment 5

In table 1 in line Age (50-59) a decimal is missing (30.0%).

Comment 6

Line 107: ATC code A01AD06 is a code for adrenalone. A code for acetylsalicylic acid is A01AD05. But you can find acetylsalicylic acid under different codes in ATC/DDD Index with different defined daily doses. The code you chose is in the group of A01 STOMATOLOGICAL PREPARATIONS. My question here is if this is some kind of mistake? The other groups with acetylsalicylic acid are B01 ANTITHROMBOTIC AGENTS (B01AC06) and N02 ANALGESICS (N02BA01). The DDD for acetylsalicylic acid is different in each code.

Comment 7

Line 118: Please also add rationale for this definition of exposure to aspirin.

Line 119: “The minimum number of DDD was >30 consumed daily.” Unit is missing (mg?).

Line 120: “…exposition to aspirin…”  Do you mean exposure to aspirin?

Lines 130 – 131: Is BMI missing?

Comment 8

In this study data from different data bases are used. Please describe in the Methods chapter the data linkage procedure.

Comment 9

In the Results chapter there are some issues with compatibility of results in text and tables.

Are results from lines 153 to 160 from the Table 2? If yes, some results do not match.

Table 2 and table 3 have the same title.

Are results from lines 161 to 169 from the Table 3? If yes, some results do not match.

Line 169: It is not specified in the text what Figure 3 shows.

There is something wrong with table 4. Please check the table. It looks like all the results should be moved two rows down.

Line 181: It is not specified in the text what Figure 4 shows.

Comment 10

Lines 273 – 276: These results are not presented. There is no Supplementary table.

Comment 11

Due to the title of this article and objective of this study I would be also interested to see analysis stratified by aspirin use.

Comment 12

The format of the references should be consistent with journal instructions:

https://www.mdpi.com/journal/ijerph/instructions#preparation

https://www.mdpi.com/authors/references

Line 345: Please follow referencing instruction from the Global Cancer Observatory for their website.

Reviewer 2 Report

The paper describes the results of a cohort retrospective study among adults aged 50 years or older living in a region of Catalonia, Spain, in order to measure the association between ASA use and CRC. 

Specific comments 

1.The authors define the follow-up period between 2007 and 2016. The incidence of CRC cases is measured between 2012 and 2016  

In my opinion, is not clear when the baseline measure of the factors studied, 2007?, occurs. Clarify. On the other hand, why are CRC cases was measured between 2012 and 2016 and not at the end of the period (2016?) Clarify. 

2. Authors should define and explain how CRC cases were identified in the initial cohort, how it was performed, and how many presented CRC cases were removed from follow-up. 

3. The baseline study risk factors (tobacco, obesity, alcohol consumption and ASA) were obtained from electronic records. Did they exist in 2007? 

4. Table 1. Present it stratified by gender with comparative statistical analysis.

 5. The prevalence of tobacco use (Table 1) is well below the prevalence of the country (9% for the study and 26% in the Spanish population over 15 years of age). This point should be discussed in the limitations of the study into the conclusions section. 

6. In the statistical analysis section, authors should included whether they are using cumulative incidence (cases/initial cohort) or incidence rate (cases/observation time) for CRC. In tables and graphs they should be shown as rates (table 2 specify that this is an incidence rate). 

7. Table 3. Title "Multivariate analysis.....". At the bottom of the table specify. By which variables it is adjusted. 

8. Table 4. Title: "Adjusted Cox Regression model....". From my point of view in the (stratified) table should not include HR by gender. Specify at the bottom of the table the variables by which it is adjusted. 

9.Discussion first paragraph (line 193) should will be follows: "Our results confirm the negative association between aspirin consumption and CRC independently of the other risk factors measured."   

10. In the conclusions section, the limitations of the study should be included, some of them discussed in previous points. Also, the limitations for the study not to measure some variables continuously such as tobacco consumption, for example and food and dietary factors not included in the study with an important role in CRC. 

11. Finally, the type of CRC (familial polyposis, familial cancer, non-familial cancer, etc.) should be presented (in results). If this is not possible, include this possible bias in limitations. 

Round 2

Reviewer 1 Report

All comments from first review report were sufficiently addressed. I have no further comments.